

# HCGAN: hierarchical contrast generative adversarial network for unpaired sketch face synthesis

Kangning Du, Zhen Wang, Lin Cao, Yanan Guo, Shu Tian and Fan Zhang

School of Information and Communication Engineering, Beijing Information Science and Technology University, Key Laboratory of Information and Communication Systems, Ministry of Information Industry, Beijing, China

## ABSTRACT

Transforming optical facial images into sketches while preserving realism and facial features poses a significant challenge. The current methods that rely on paired training data are costly and resource-intensive. Furthermore, they often fail to capture the intricate features of faces, resulting in substandard sketch generation. To address these challenges, we propose the novel hierarchical contrast generative adversarial network (HCGAN). Firstly, HCGAN consists of a global sketch synthesis module that generates sketches with well-defined global features and a local sketch refinement module that enhances the ability to extract features in critical areas. Secondly, we introduce local refinement loss based on the local sketch refinement module, refining sketches at a granular level. Finally, we propose an association strategy called "warmup-epoch" and local consistency loss between the two modules to ensure HCGAN is effectively optimized. Evaluations of the CUFS and SKSF-A datasets demonstrate that our method produces high-quality sketches and outperforms existing state-of-the-art methods in terms of fidelity and realism. Compared to the current state-of-the-art methods, HCGAN reduces FID by 12.6941, 4.9124, and 9.0316 on three datasets of CUFS, respectively, and by 7.4679 on the SKSF-A dataset. Additionally, it obtained optimal scores for content fidelity (CF), global effects (GE), and local patterns (LP). The proposed HCGAN model provides a promising solution for realistic sketch synthesis under unpaired data training.

# INTRODUCTION

Unpaired sketch face synthesis transforms optical face images into sketches without paired training data. This technique shares similarities with image style transfer (*Lin, Pang & Xia, 2020*; *Zhu et al., 2017*; *Park et al., 2020*) and has implications in entertainment and criminal investigations. Generating paired high-quality images for sketch face synthesis is often time-consuming and expensive, requiring skilled artists to create them by hand using traditional methods. Therefore, unpaired sketch face synthesis presents a more practical solution when acquiring paired data is prohibitively time-consuming and expensive.

Corresponding author
Lin Cao, charlin26@163.com

Nevertheless, sketch face synthesis from unpaired data is a more intricate task compared to using paired data (*Yi et al., 2020a*). The imperfect pixel correlation between unpaired sketches and optical images poses challenges in capturing local details and may lead to feature displacement and shadowing issues. Existing methods often struggle to address shadow artifacts at local edges, significantly impacting the realism and accuracy of facial details.

In the past decade, various methods have been explored for sketch face synthesis, including traditional Bayesian inference (*Chen et al., 2001*; *Nefian & Hayes, 1999*; *Wang et al., 2013*; *Peng et al., 2015*), representation learning (*Tang & Wang, 2002*; *Liu et al., 2005*), and subspace learning (*Huang & Wang, 2013*; *Ji et al., 2011*; *Wang et al., 2012*; *Gao et al., 2012*; *Zhang et al., 2011*). However, these methods require extensive computational resources and training data, hindering real-time performance and limit practicality in real-life applications.

Recently, end-to-end generative adversarial networks (GANs) have gained traction in sketch face synthesis. For example, *Yi et al. (2019)* and *Yi et al. (2020b)* proposed APDrawing and APDrawing++, which divided images into sections, transformed each into a partial sketch, and then fused these sketches to generate a comprehensive pseudo-sketch. However, these methods neglected the relevant parameters of each individual section during global network training. Moreover, they required paired training data, which is not only expensive but also difficult to procure, thereby restricting their ability to adapt to various lighting conditions in unconstrained or natural environments. Within the realm of unpaired image style transfer, *Zhu et al. (2017)* proposed Cycle-GAN which introduces a cycle consistency loss that employs two generators to process the two inputs and reconstruct images. Other methods (*Lin, Pang & Xia, 2020*; *Taigman, Polyak & Wolf, 2017*; *Bousmalis et al., 2017*) adopt a similar two-way and double-branch structure. However, these methods often indiscriminately embed invisible reconstruction information across the entire sketch, consequently reducing the quality of the generated sketch and leading to the partial absence of crucial facial features. To address this issue, *Chen et al. (2023)* proposed a semi-supervised approach with a noise-injection strategy named Semi-Cycle-GAN (SCG), which reduces the impact of salt-and-pepper noise on the synthesized sketch. *Park et al. (2020)* introduced contrastive unpaired translation (CUT), and *Gou et al. (2023)* proposed multi-feature contrastive learning (MCL). However, these methods produce sketches with local details that are not sufficiently realistic, leading to issues such as feature displacement and shadowing.

While the aforementioned methods tolerate certain reconstruction quality shortcomings and meet the general appearance requirements of sketch face synthesis, they struggle to preserve the integrity and realism of facial details. In response to this challenge, this article presents the hierarchical contrast generative adversarial network (HCGAN), a novel approach designed to generate high-quality facial sketches with precise local details from unpaired input data. HCGAN is comprised of two main modules: the global sketch synthesis module and the local sketch refinement module. We also introduce an association strategy called "warmup-epoch" to optimize the synthesis process to establish the necessary connections between the two modules.

Specifically, HCGAN comprises two distinct stages, separated by the "warmup-epoch" association strategy. In the initial stage, we introduce a global sketch synthesis module and a local sketch refinement module to produce a high-quality overall sketch and enhance the extraction of local features. In the subsequent stage, we incorporate local consistency loss and local refinement loss to optimize the global sketch synthesis module in collaboration with the local sketch refinement module. This optimization process yields sketches with superior local quality.

Our main contributions can be summarized as follows:

- Novel HCGAN approach: We present a novel HCGAN for unpaired face-to-sketch transformation that overcomes the limitations of existing approaches relying on paired data. It features a global sketch synthesis module for macroscopic-level sketch generation and a local sketch refinement module for extracting local features.

- Local refinement loss: To enhance the control of the global sketch synthesis module over the details of the synthesized sketch at local-level, we propose the local refinement loss based on the local sketch refinement module. It act on the global sketch synthesis module to enhance the realism of synthesized sketch details, ensuring the synthesized sketch exhibits more accurate local feature expression.

- Effective association strategy: To ensure efficient optimization and the production of high-quality sketches, we propose an effective association strategy between the global sketch synthesis module and the local sketch refinement module called the "warmup-epoch". Simultaneously, we employ the local consistency loss to establish a close relationship and optimize between the two modules.

- Experimental results on the CUFS and SKSF-A datasets demonstrate the superiority of HCGAN in synthesizing highly realistic sketches with intricate local details, outperforming existing methods when using unpaired inputs.

# RELATED WORKS

Sketch face synthesis is a challenging task that differs from image style transfer due to the sensitivity of sketches to misplaced or missing lines. Traditional methods heavily rely on the quality of the training dataset and often prove ineffective for practical applications. In contrast, deep learning-based methods can balance the influence of individual data in the training dataset, yet they frequently yield results with reduced clarity and absent facial features. While GANs have shown promise in addressing local deformation problems, their performance in training with unpaired data remains unsatisfactory. As a result, there is an urgent need for a novel approach capable of synthesizing high-quality sketches with detailed local features from unpaired inputs.

## Traditional methods

Traditional sketch synthesis approaches aim to learn the mapping relationship between images and sketches. This method mainly includes Bayesian inference, representation learning, and subspace learning models.

Bayesian inference models update the sketch localization based on a probabilistic model using real data. For example, *Chen et al. (2001)* proposed an instance-based sketch face synthesis system that utilizes a nonparametric sampling algorithm to learn sketching style details. *Nefian & Hayes (1999)* used an embedded Hidden Markov model to capture the nonlinear relationship between picture and sketch pairs. *Wang et al. (2013)* introduced a transduction learning method for synthetic sketching, employing a dynamic process to minimize the loss of a given test sample. Furthermore, *Peng et al. (2015)* devised a superpixel approach based on Markov models to enhance flexibility without dividing photos into regular rectangular blocks. However, these traditional methods often produce suboptimal results, partly because they rely on manually crafted features and do not always capture the complexity of the mapping relationship.

Subspace learning models aim to transform high-dimensional spatial features into low-dimensional ones. *Tang & Wang (2002)* proposed a series of example-based methods based on linear feature transformation techniques. However, these methods rely on global linear systems and fail to capture the complex relationship between photo and sketch pairs fully. To address this issue, *Liu et al. (2005)* used the local linear embedding (LLE) method, which ensures locally geometrically similar stream shapes for photo and sketch patches in two distinct image spaces. Nevertheless, this method separates pseudo-sketch generation and representation learning into two distinct processes, resulting in suboptimal outcomes.

*Huang & Wang (2013)* introduced a joint learning framework encompassing domain-specific dictionary learning and subspace learning. The representation learning model primarily relies on sparse coding with dictionary learning. *Ji et al. (2011)* highlighted the limitations of capturing personalized features through a synthetic process. *Wang et al. (2012)* proposed a semi-coupled dictionary learning method, employing a linear transformation to bridge the gap between two different domain-specific representations. *Gao et al. (2012)*, building on a two-step algorithm. *Zhang et al. (2011)*, presented a selection scheme for generating initial pseudo-images and introduced sparse representation-based enhancement (SRE) for sketch synthesis.

However, these methods involve extensive calculations, resulting in poor real-time performance and limited practicality. Moreover, they rely on large-scale training data, making achieving satisfactory results with limited training data challenging.

## Deep learning-based methods

Deep learning-based methods have become increasingly popular for sketch synthesis due to their ability to train on datasets and generate high-quality results. Among these methods, generative adversarial networks (GANs) are widely used.

*Zhang et al. (2015)* introduced the first deep photo-sketch synthesis model using a fully convolutional neural network (FCNN), but struggled to preserve certain details. To mitigate this issue, *Zhang et al. (2018a)* proposed PGAN, which uses a specific parametric Sigmoid activation function to reduce the impact of photo prior and lighting variations. Similarly, *Wang & Sindagi (2018)* introduced a synthesis method called PS2MAN, which employs two U-Net architectures within a multiple adversarial network to generate high-quality images at varying resolutions. To address blurring and distortion,

*Zhang et al. (2018b)* proposed a multi-domain adversarial learning (MDAL) approach to sketch face synthesis. *Liang et al. (2023)* proposed Parallel Multistage GANs for Face Image Translation (PMSGAN), achieving synthesis from coarse to fine.

More and more methods are introduced of identity-aware models (*Fang et al., 2020*; *Lin et al., 2020*), incorporating novel perceptual losses to train image generation models with facial recognition as the ultimate goal. Other notable methods include *Yu et al. (2020)*, who introduced a synthesis-assisted generative adversarial network utilizing facial synthesis information, and *Duan et al. (2020)*, who implemented a multiscale self-attentive residual learning framework for face photo-sketch conversion. Another notable approach (*Goodfellow et al., 2020*) does not require any source domain images for training and utilizes deep features extracted from CNNs and manual features.

Moreover, researchers increasingly emphasize the importance of preserving the content features of the input optical image while modifying stylistic features in sketch-based face synthesis. *Seo, Ashtari & Noh (2023)* introduced a style encoder based on Cycle-GAN to refine the network's control over styles. *Cui et al. (2021)* proposed the Self-Supervised Semantic Network (SSNet), incorporating both style and semantic feature encoding. *Gao et al. (2021)* introduced CHAN, reinforcing the network's perceptual capabilities for features such as textures. *Li et al. (2022)* introduced color refinement loss and texture loss, further augmenting the network's control over content features. *Kong et al. (2023)* proposed an Asymmetric Double-Stream Generative Adversarial Network (ADS-GAN), incorporating edge regularization constraints. *Yun et al. (2024)* proposed a novel sketch face synthesis method called StyleSketch, based on prior knowledge. It utilizes pre-trained StyleGAN to extract rich semantic deep features, achieving satisfactory results.

It is worth mentioning that diffusion (*Sohl-Dickstein et al., 2015*), a newer and better general style transfer model, is currently available. Diffusion models is to find a noise map and a conditioning vector corresponding to a generated image. It is a potential way to improve the quality of example-guided artistic image generation. *Dhariwal & Nichol (2021)* invert the deterministic DDIM (*Song, Meng & Ermon, 2020*) sampling process in closed form to obtain a latent noise map that will produce a given real image. *Ramesh et al. (2022)* develop a text-conditional image generator based on the diffusion models and the inverted CLIP. The above methods are difficult to generate new instances of a given example while maintaining fidelity. *Zhang et al. (2023)* propose an inversion-based style transfer method (InST), which can efficiently and accurately learn the key information of an image, thus capturing and transferring the artistic style of a painting. However, the above diffusion-based methods cannot convert optical images into sketch images when the training set size is small. The reason is that when the training data is small, it is difficult for them to correct inappropriate weights in the pre-training weights, making it difficult to ensure that the style features of the optical image are completely removed, resulting in the failure of the sketch face synthesis task.

Detailed sketches can only be drawn manually by painters, resulting in sketched face datasets that are often small in size. When training data is small, GAN-based methods are often the most effective. In addition, compared with the Diffusion-based method, the GAN-based method has the advantages of small parameter size and fast optimization.

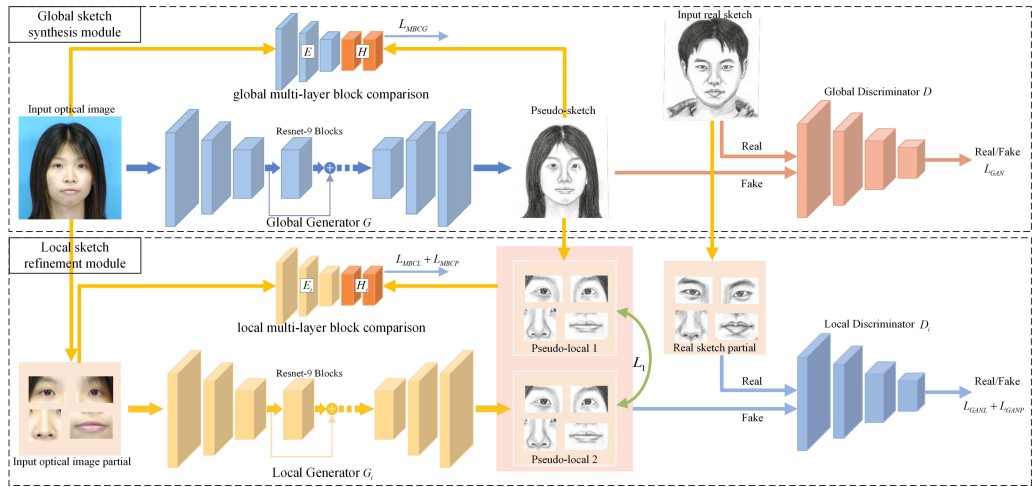

**Figure 1  Framework of hierarchical contrast generative adversarial network (HCGAN).** It comprises a global sketch synthesis module and a local sketch refinement module. The former generates a sketch, while the latter optimizes the local refinement loss to optimize the global generator further. To establish an effective association optimization between the two modules, the HCGAN implements the local consistency loss and adopts the "warmup-epoch" strategy to adjust each loss coefficient during training.

However, although the above GAN-based methods have synthesized high-quality sketches, local details are still unsatisfactory, and the dependence on paired training data limits the broadening of application scenarios of most methods.

# PROPOSED METHOD

## Overview

Our article introduces a two-module generative adversarial network framework, as depicted in Fig. 1, incorporating novel loss functions for training a network $F(\cdot)$ to produce high-quality sketches from input images. The framework comprises a global sketch synthesis module and a local sketch refinement module, each equipped with dedicated generators, discriminators, and block feature extractors.

The global module generates overall sketches, while the local module focuses on extracting accurate local features. However, solely relying on adversarial loss neglects image details, which becomes evident in unpaired training data scenarios, resulting in pseudo-sketch faces with insufficient local details. To address this limitation, we propose multi-layer block contrast loss, local consistency loss, and local refinement loss to generate refined local details in the sketch. This enhancement leads to pseudo-sketches with more precise details. Moreover, the framework can be trained end-to-end without requiring post-processing.

In our article, we refer to the four local generators, discriminators, and block feature extractors as $G_i$, $D_i$, and $H_i$, respectively.

## Global sketch synthesis module

The global sketch synthesis module mainly consists global generator $G$, global discriminator $D$, and global block feature extractor $H$. The block feature extractor $H$ and the encoding part $E$ of the global generator $G$ form the global multi-layer block comparison module. The global sketch synthesis module takes the unpaired image pair $\{P_i, S_j\}$ as input. The optical face image $P_i$ is passed through the global generator $G$ to obtain pseudo-sketch $S'_i = G(P_i)$. Meanwhile the true sketch $S_j$ and the pseudo sketch $S'_i$ are input to the global discriminator $D$ to assess the authenticity of the pseudo sketch $S'_i$.

The generator $G$ used in this article is based on the generator in *Johnson, Alahi & Fei-Fei (2016)* and includes residual blocks in its middle layer. This mitigates the problem of network degradation that occurs when increasing the number of network layers. Incorporating residual blocks reduces the loss of crucial features and improves the effectiveness of network training, resulting in the generation of more realistic and high-quality sketches.

The discriminator $D$ used in this article is based on the discriminator in *Isola et al. (2017)*. This discriminator focuses solely on the local structure of the image, allowing it to effectively learn the high-frequency information and detailed features of the image. This approach reduces the parameters required during training and improves training efficiency.

The block feature extractor $H$ in HCGAN is based on the architecture of the block feature extractor used in SimCLR (*Chen et al., 2020*), which includes two fully connected layers. After the encoder extracts features from the optical or sketch image, the block feature extractor $H$ stacks these features in blocks, creating conditions for subsequent multi-layer block contrast loss.

The global sketch synthesis module utilizes global adversarial loss and global multi-layer block contrast loss for optimization. Additionally, in the latter stage of training, it introduces local refinement loss (as mentioned in 'Local refinement loss') to enhance the optimization process further.

**Global adversarial loss** $L_{GAN}$ aims to synthesize sketches with the sketch style under unpaired input. It establishes a "gaming" optimization process between the global generator $G$ and the global discriminator $D$, enabling the global generator $G$ to fully learn the input sketch style and enhance the composite image's quality. The equation of $L_{GAN}$ is as follows.

$$L_{GAN}(G, D, P, S) = E_{P_i \sim P} log(1 - D(G(P_i))) + E_{S_j \sim S} \log D(S_j) \tag{1}$$

**Global multi-layer block contrast loss** $L_{MBCG}$ is inspired by the literature *Park et al. (2020)* to maximize the mutual information between the output sketch and the input image. It could be helpful to improve the composite image's quality. This loss is similar to the local multi-layer block contrast loss $L_{MBCL}$ in Eq. (6) and multi-layer refinement loss $L_{MBCP}$ in Eq. (11). They are all based on multi-layer block contrast loss.

The loss of multi-layer block comparison is presented in Fig. 2, which corresponds to both the global and the local multi-layer block comparisons as shown in Fig. 1. We obtain an image block (or a query block) from the encoding feature of each layer of the

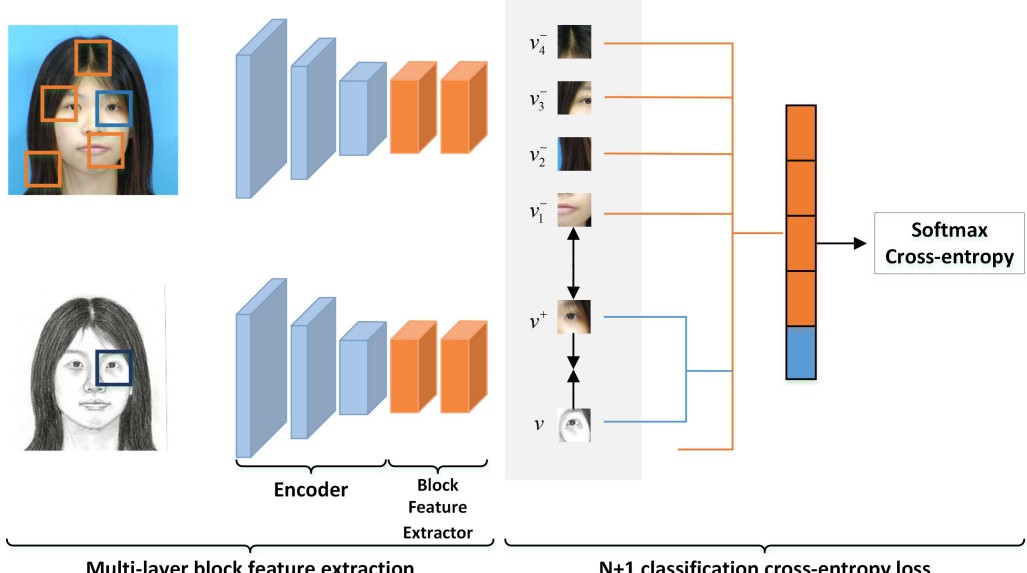

**Figure 2** **Multi-layer block contrast loss in $L_{MBCG}$.** Similarly, both $L_{MBCL}$ and $L_{MBCP}$ work similarly.

pseudo-sketch. From the input optical image, we select an image block (or a positive class block) with the same position as the coding feature of each layer. The rest of the image blocks, which have different positions from the coding features of each layer, are considered negative class blocks. The query block is expected to have a strong association with the positive class block and a minor association with the negative ones. To construct N+1 classification, we crop N negative class blocks at different positions of the input optical image paired with a positive class block. We employ the Cross-Entropy loss to maximize the association between the query block and the positive class block. The loss is formulated as follows.

$$l\left(v, v^+, v^-\right) = -\log\left[\frac{exp\left(v \bullet v^+/\tau\right)}{exp\left(v \bullet v^+/\tau\right) + \sum_{n=1}^{N} exp\left(v \bullet v^-/\tau\right)}\right] \tag{2}$$

where, v denotes the query block, $v^+$ denotes the positive class block, $v^-$ denotes the negative class block, and $\tau$ denotes the scaling factor mentioned in *Park et al. (2020)*. Specifically, we set $\tau$ to 0.07 in the experiments conducted in this article.

The global multi-layer block contrast module provides the global multi-layer block contrast loss. To compute this loss, we feed the input optical image $P_i$ and the output pseudo-sketch image $S_i'$ into the encoder $E$. We then use the output of the middle layer of the encoder as the input to the block feature extractor $H$. The block feature extractor $H$ chunks the input, creating a classification problem for each layer, and calculates the Cross-Entropy loss to obtain the global multi-layer block contrast loss, denoted by $L_{MBCG}$.

$$L_{MBCG}(G, H, P) = E_{P_i \sim P} \sum_{m=1}^{M} \sum_{n=1}^{N+1} l\left(\hat{x}_m^n, x_m^n, x_m^{N-}\right) \tag{3}$$

where,

$$\hat{x}_m^n = \left\{ H\left( E_m\left( S_i' \right) \right) \right\}_n = \{ H\left( E_m\left( G(P_i) \right) \right) \}_n \tag{4}$$

where, $E_m\left( S_i' \right)$ represents the output of m-th layer of the pseudo-sketch obtained *via* the encoder $E$. Similarly, $\hat{x}_m^n$ denotes the n-th block, which is obtained by chunking $E_m\left( S_i' \right)$ using the block feature extractor $H$. On the other hand, $x_m^n = \{H\left( E_m(P_i) \right)\}_n$ implies that the m-th layer output of the input optical image is passed through the encoder $E$, and then chunked *via* the block feature extractor $H$ to yield the positive class block (with sequence number $n$). $x_m^{N-} = \{H\left( E_m(P_i) \right)\}_{!n}$ denotes the blocks obtained after dividing the output of the m-th layer of encoder $E$ of the input optical image *via* the block feature extraction network $H$. These blocks exclude the positive class blocks (negative class blocks), as identified by the symbol "!" in "!$n$". The symbol "!$n$" indicates the negation of "$n$", *i.e.,* all blocks except the one with the sequence number $n$. The Cross-Entropy loss operation described in Eq. (2) is represented by $l$.

### Local sketch refinement module

The local sketch refinement module comprises four generators $G_i$, four discriminators $D_i$, and four local block feature extractors $H_i$. It takes in four local regions extracted from the optical image $P_i$, the sketch image $S_j$, and the pseudo-sketch $S_i'$. The main objective of this module is to provide adversarial refinement loss $L_{GANP}$, multi-layer refinement loss $L_{MBCP}$, and local consistency loss $L_1$. These three losses help the global generator $G$ produce more locally refined pseudo-sketches $S_i'$.

The local multi-layer block comparison module uses the encoding part $E_i$ of the local generator $G_i$ as its encoder. To obtain more targeted features for different facial parts, we trained four local generators, denoted by $G_i$. Specifically, each local generator $G_i$ accepts one of four parts of the optical image $P_i$: left eye $P_{iel}$, right eye $P_{ier}$, nose $P_{in}$, and mouth $P_{im}$. It then generates the respective local pseudo-sketches $S_{iel}'$, $S_{ier}'$, $S_{in}'$ and $S_{im}'$ for each region.

To improve the performance of the local multi-layer block comparison module and the local discriminator, we optimize them using two loss functions: the local adversarial loss and the local multi-layer block contrast loss. These loss functions aim to generate more precise and visually appealing local pseudo-sketches with the local generator $G_i$. Conversely, the role of the local discriminator $D_i$ is to differentiate between real and generated local sketches. By optimizing these components, we aim to enhance the module's ability to extract local features.

**Local adversarial loss** $L_{GANL}$ is based on adversarial loss. We utilize a competitive optimization approach between the local generator and local discriminator using local adversarial loss to enhance the local feature encoding of the encoding part of the local generator. This method helps us to obtain optimized local generator encoding parts and

local discriminators, which are essential for the local refinement loss.

$$
\begin{aligned}
L_{GANL}(G_i, D_i, P, S) = {} & E_{P_{iel} \sim P} log \left(1 - D_{iel}\left(G(P_{iel})\right)\right) + E_{S_{jel} \sim S} log D_{iel}\left(S_{jel}\right) \\
& + E_{P_{ier} \sim P} log \left(1 - D_{ier}\left(G(P_{ier})\right)\right) + E_{S_{jer} \sim S} log D_{ier}\left(S_{jer}\right) \\
& + E_{P_{in} \sim P} log \left(1 - D_{in}\left(G(P_{in})\right)\right) + E_{S_{jn} \sim S} log D_{in}\left(S_{jn}\right) \\
& + E_{P_{im} \sim P} log \left(1 - D_{im}\left(G(P_{im})\right)\right) + E_{S_{jm} \sim S} log D_{im}\left(S_{jm}\right)
\end{aligned}
\tag{5}
$$

**Local multi-layer block contrast loss** $L_{MBCL}$ is similar to $L_{MBCG}$ in Eq. (3). It is particularly useful for improving the performance of the local generator by enhancing the correlation between the input optical local feature and the output sketch local feature. This improvement enables a better encoding part for the local generator, facilitating enhanced capture and encoding of local features.

$$
\begin{aligned}
L_{MBCL}(G_i, H_i, P) = {} & E_{P_{in} \sim P} \sum_{m=1}^{M} \sum_{n=1}^{N+1} l\left(\hat{y}_{min}^n, y_{min}^n, y_{min}^{N-}\right) + E_{P_{im} \sim P} \sum_{m=1}^{M} \sum_{n=1}^{N+1} l\left(\hat{y}_{mim}^n, y_{mim}^n, y_{mim}^{N-}\right) \\
& + E_{P_{ier} \sim P} \sum_{m=1}^{M} \sum_{n=1}^{N+1} l\left(\hat{y}_{mier}^n, y_{mier}^n, y_{mier}^{N-}\right) + E_{P_{iel} \sim P} \sum_{m=1}^{M} \sum_{n=1}^{N+1} l\left(\hat{y}_{miel}^n, y_{mie}^n, y_{mie}^{N-}\right).
\end{aligned}
\tag{6}
$$

As with the global multi-layer block contrast loss $L_{MBCG}$, we obtain $y_{min}^n$ for the nose part $P_{in}$ by consecutively feeding the input through the local generator $G_{in}$ and the nose local block feature extractor $H_{in}$. The equation is as follows.

$$
\hat{y}_{min}^n = \left\{H_{in}\left(E_{inm}\left(S_{in}'\right)\right)\right\}_n = \left\{H_{in}\left(E_{inm}\left(G_{in}\left(P_{in}\right)\right)\right)\right\}_n
\tag{7}
$$

Moreover:

$$
\begin{aligned}
y_{min}^n &= \left\{H_{in}\left(E_{inm}\left(P_{in}\right)\right)\right\}_n \\
y_{min}^{N-} &= \left\{H_{in}\left(E_{inm}\left(P_{in}\right)\right)\right\}_{!n}
\end{aligned}
\tag{8}
$$

Similarly, the respective characters in multi-layer refinement loss $L_{MBCP}$ in Eq. (11) can be obtained.

$$
\begin{aligned}
\hat{z}_{min}^n &= \left\{H_{in}\left(E_{inm}\left(S_{iin}'\right)\right)\right\}_n \\
z_{min}^n &= \left\{H_{in}\left(E_{inm}\left(P_{in}\right)\right)\right\}_n \\
z_{min}^{N-} &= \left\{H_{in}\left(E_{inm}\left(P_{in}\right)\right)\right\}_{!n}
\end{aligned}
\tag{9}
$$

**Local refinement loss** operates on the global sketch synthesis module and is constrained by both multi-layer features and realism, ensuring the synthesized sketch exhibits more accurate local feature expression. It includes multi-layer refinement loss $L_{MBCP}$ and adversarial refinement loss $L_{GANP}$ as follows.

**Adversarial refinement loss** $L_{GANP}$ is optimized between the global generator and the local discriminators. We optimize the details of sketches generated by the global generator using the global sketch synthesis module.

$$
\begin{aligned}
L_{GANP}(G_i, D_i, P, S) = {} & E_{P_{iel} \sim P} log \left(1 - D_{iel}\left(S_{iiel}'\right)\right) + E_{S_{iel} \sim S} log D_{iel}\left(S_{jel}\right) \\
& + E_{P_{ier} \sim P} log \left(1 - D_{ier}\left(S_{iier}'\right)\right) + E_{S_{ier} \sim S} log D_{ier}\left(S_{jer}\right) \\
& + E_{P_{in} \sim P} log \left(1 - D_{in}\left(S_{iin}'\right)\right) + E_{S_{in} \sim S} log D_{in}\left(S_{jn}\right) \\
& + E_{P_{im} \sim P} log \left(1 - D_{im}\left(S_{iim}'\right)\right) + E_{S_{im} \sim S} log D_{im}\left(S_{jm}\right)
\end{aligned}
\tag{10}
$$

**Multi-layer refinement loss** $L_{MBCP}$ is introduced to constrain the association between the parts of the generated sketches and the corresponding input. This loss is necessary to prevent the loss of crucial local information in the generated sketches.

$$
\begin{aligned}
L_{MBCP}(G_i, H_i, P) = {} & E_{P_{in}\sim P, S'_{iin}\sim S'_i} \sum_{m=1}^{M}\sum_{n=1}^{N+1} l\left(\hat{z}^n_{min}, z^n_{min}, z^{N-}_{min}\right) \\
& + E_{P_{im}\sim P, S'_{iim}\sim S'_i} \sum_{m=1}^{M}\sum_{n=1}^{N+1} l\left(\hat{z}^n_{mim}, z^n_{mim}, z^{N-}_{mim}\right) \\
& + E_{P_{ier}\sim P, S'_{iier}\sim S'_i} \sum_{m=1}^{M}\sum_{n=1}^{N+1} l\left(\hat{z}^n_{mier}, z^n_{mier}, z^{N-}_{mier}\right) \\
& + E_{P_{iel}\sim P, S'_{iiel}\sim S'_i} \sum_{m=1}^{M}\sum_{n=1}^{N+1} l\left(\hat{z}^n_{miel}, z^n_{miel}, z^{N-}_{mierl}\right)
\end{aligned}
\tag{11}
$$

## Association strategy

Optimization balancing is a recurring challenge in multi-module networks, including HCGAN, which consists of a global sketch synthesis module and a local sketch refinement module. Additionally, HCGAN involves local refinement loss between two modules, which poses a challenge for optimization balancing. To deal with this challenge, this article proposes two methods: local consistency loss and "warmup-epoch" strategy.

**Local consistency loss** $L_1$ ensures that the correlation between the two modules is optimized and addresses the issue of significant differences between the locally generated sketches by the global generator $G$ and the local generator module $G_i$. This discrepancy reduces the effectiveness of $L_{MBCP}$ and $L_{GANP}$ in optimizing the global generator $G$.

$$
L_1 = \left\| S'_{iel} - S'_{iiel} \right\| + \left\| S'_{ier} - S'_{iier} \right\| + \left\| S'_{in} - S'_{iin} \right\| + \left\| S'_{im} - S'_{iim} \right\|
\tag{12}
$$

The **"warmup-epoch" strategy** optimizes the relation between local feature extraction and global optimization and is inspired by an approach described in the literature (*Wu et al., 2020*; *Lyu, Rosin & Lai, 2023*). The proposed model utilizes a global sketch synthesis module to generate pseudo-sketches. Additionally, the local sketch refinement module provides local refinement loss to enhance local details. By implementing the "warmup-epoch" strategy, the local sketch refinement and global sketch synthesis modules are initially optimized separately. This allows the global sketch synthesis module to generate higher-quality global sketches, enabling the local sketch refinement module to distill local features better. When an improved global sketch is generated, the multi-layer refinement loss and adversarial refinement loss provided by the local sketch refinement component are employed to optimize the global generator $G$ for local detailing.

The total loss function of the network is:

$$
L_{total} = a \times L_{GANL} + b \times L_{MBCL} + c \times L_{GANP} + d \times L_{MBCP} + e \times L_{GAN} + f \times L_{MBCG} + L_1.
\tag{13}
$$

According to the "warmup-epoch", we divide the training process into two stages. In the first stage of this experiment, $c = d = 0, a = b = 0.25, e = f = 1$. In the second stage

of the training, $a = b = c = d = 0.25$, $e = f = 1$. It is worth mentioning that, given that sketches, as facial images, require particular attention to the coordination among different components at the global-level, a higher weight is allocated to the loss associated with global sketching in the overall loss function. In comparison, a lower weight is assigned to the loss linked to local refinement to avoid excessive local refinement leading to overall imbalance.

# EXPERIMENT

Due to the high cost of acquiring high-quality sketch images, which can only be achieved through manual drawing, datasets are generally small. To simulate realistic scenarios with insufficient high-quality data and fully reflect the robustness of HCGAN, ablation experiments, and comparison experiments are carried out on the CUFS dataset (*Wang & Tang, 2008*) and the SKSF-A dataset (*Yun et al., 2024*) in this section.

## Evaluation metrics

Currently, many metrics are used for image quality assessment. To better simulate human visual perception, we choose to use Fréchet Inception Distance (FID) (*Heusel et al., 2017*) along with Content Fidelity (CF), Global Effects (GE), and Local Patterns (LP) (*Wang et al., 2021*) to evaluate the test results.

FID is a metric to assess the quality of images generated by generative models. It is calculated by comparing the feature distributions of generated images with those of real images. Specifically, FID utilizes a deep neural network (typically a pre-trained Inception network) to extract features from images and then computes the Fréchet distance between the feature distributions of generated and real images. This distance measures the similarity between the two feature distributions.

The combination of CF, GE, and LP can effectively and comprehensively evaluate the quality of style transfer generated images. Leveraging feature extraction networks, they simulate human perception of images by focusing on content, global, and local aspects, respectively. GE emphasizes global aspects such as global colors(GC) and holistic textures(HT). Similarly, LP consists of two parts: one is to assess the similarity of local pattern counterparts directly, and the other is to compare the diversity of retrieved pattern categories.

Overall, by combining the evaluation metrics of FID, CF, GE and LP. We can better simulate human visual perception and comprehensively evaluate the quality of generated images, thus providing important guidance and reference for improving image generation technology.

## Dataset and setting

Since high-quality sketches can only be produced by hand, obtaining them is expensive and time-consuming. Therefore, current high-quality sketch datasets tend to be smaller in size. In order to fully demonstrate the effectiveness and robustness of HCGAN, the experiment section employs the CUFS and SKSF-A datasets.

The CUFS datasets comprise the CUHK, AR, and XM2VTS datasets. CUHK and AR datasets have fewer accessories, and both of them are frontal photos. There are many

**Table 1  Training and testing set division for datasets.**

| Dataset | Train | Test |
|---|---|---|
| CUHK | 88 | 100 |
| AR | 80 | 43 |
| XM2VTS | 100 | 195 |
| SKSF-A | 88 | 46 |

accessories, such as glasses and earrings, in the XM2VTS dataset. Each dataset contains a varying number of image-sketch pairs, with 188, 123, and 295 pairs, respectively. Image-sketch pairs consist of an optical photograph paired with a corresponding sketch hand-drawn by painters. The SKSF-A dataset provides 134 optical images along with corresponding sketches in seven different styles. And there is a large difference in angle. However, most of these styles are simple sketches or initial drafts, which are not suitable for training a fine-sketch synthesis model. Therefore, we only selected the style with the highest sketch quality for our experiments. The training and test sets include different numbers of optical image-sketch pairs for each dataset. The division of training and testing sets for each dataset is shown in Table 1.

Before the experiment, it is necessary to process the data. The face images and corresponding sketches from the CUFS dataset were aligned by the Multi-task Cascaded Convolutional Neural Networks (MTCNN). Furthermore, the optical images in the SKSF-A dataset featured diverse backgrounds, which could negatively impact the synthesis outcomes. Thus, we employed masks provided by the SKSF-A dataset to crop the images, replacing the backgrounds with white.

We conducted training and testing on a server equipped with an Intel(R) Xeon(R) CPU E5-2640 v4 and an NVIDIA GeForce RTX 4090. During training, a minimum of 6GB of GPU memory was required. The network is trained with a batch size of 1. The training optimizer is the Adam optimizer, and the momentum parameters $\beta_1$ and $\beta_2$ are set to 0.9 and 0.999, respectively. The training period consists of 800 epochs, with the learning rate set to 0.0002 for the first 400 epochs and gradually decaying from 0.0002 to 0 for the following 400 epochs. In order to avoid the slow model training caused by too many operations, only the outputs of 0, 4, 8, 12 and 16 layers of the encoder are used in the calculation of $L_{MBCG}, L_{MBCL}$, and $L_{MBCP}$, and the number of blocks in each layer is 64.

## Ablation experiments

To verify the effectiveness of the network proposed in this article, this section removes different components from HCGAN, including the local sketch refinement module, the local consistency loss $L_1$, adversarial refinement loss $L_{GANP}$, multi-layer refinement loss $L_{MBCP}$ and "warmup-epoch" to examine their impact on the final synthesis results. Figure 3 illustrates the experimental results of method validity for different datasets, while Tables 2 and 3 compares the quantitative validity indices across different datasets.

The results are presented in Table 2 indicates that the complete model outperforms individual ablation experiments. Specifically, after adding the local sketch refinement module, the FID metric is reduced by about 26 under the CUHK dataset, 24 under the

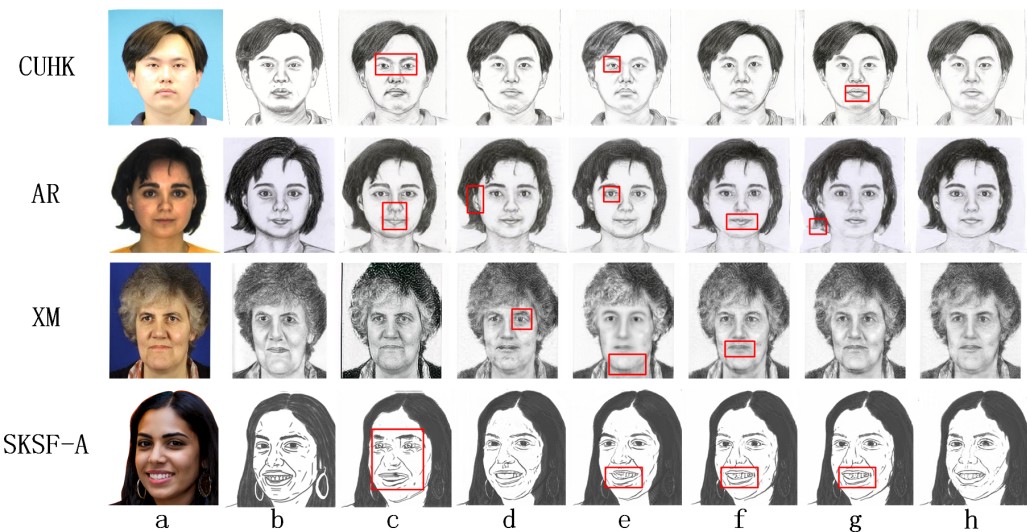

**Figure 3** **Comparison of synthetic effects of ablation experiments.** (A) Optical Image, (B) real sketch, (C) W/O LSRM, (D) W/O $L_1$, (E) W/O $L_{GANP}$, (F) W/O $L_{MBCP}$, (G) W/O "warmup-epoch", (H) HCGAN.

**Table 2** **Fréchet inception distance of ablation experiments.**

| Dataset | W/O LSRM | W/O $L_1$ | W/O $L_{GANP}$ | W/O $L_{MBCP}$ | W/O "warmup-epoch" | HCGAN |
|---|---|---|---|---|---|---|
| CUHK | 74.7018 | 63.1057 | 72.6184 | 70.4261 | 80.2496 | **48.5769** |
| AR | 88.8440 | 62.3499 | 67.5015 | 66.3910 | 65.3214 | **62.0253** |
| XM2VTS | 96.3619 | 36.9687 | 92.8553 | 51.7058 | 36.9941 | **30.6324** |
| SKSF-A | 92.9306 | 84.3543 | 89.6085 | 79.2687 | 80.7965 | **67.7272** |

Notes.
The best FID are bolded.

**Table 3** **CF, GE, and LP of ablation experiments.**

| Dataset | W/O LSRM | W/O $L_1$ | W/O $L_{GANP}$ | W/O $L_{MBCP}$ | W/O "warmup-epoch" | HCGAN |
|---|---|---|---|---|---|---|
| CUHK | **0.7818** | 0.7548 | 0.5986 | 0.6011 | 0.6020 | 0.7675 |
| | 0.7771 | 0.7934 | 0.7573 | 0.6712 | 0.7638 | **0.8169** |
| | 0.7544 | 0.7299 | 0.6827 | 0.6812 | 0.7001 | **0.7597** |
| AR | 0.8137 | 0.8202 | **0.8278** | 0.8083 | 0.7994 | 0.8235 |
| | **0.8819** | 0.8496 | 0.8139 | 0.8705 | 0.8703 | 0.8691 |
| | 0.7085 | 0.7558 | 0.7448 | 0.7678 | 0.7587 | **0.7815** |
| XM2VTS | 0.5614 | 0.6542 | 0.6638 | 0.6388 | **0.7287** | 0.7190 |
| | 0.6396 | 0.5546 | 0.6496 | 0.6156 | 0.9510 | **0.9538** |
| | 0.6033 | 0.5634 | 0.6352 | 0.6668 | 0.7313 | **0.7457** |
| SKSF-A | 0.7099 | 0.7277 | **0.7285** | 0.7044 | 0.7034 | 0.7229 |
| | 0.9799 | 0.9446 | 0.9765 | 0.9800 | **0.9849** | 0.9783 |
| | 0.6973 | 0.7760 | 0.7409 | 0.7370 | 0.7235 | **0.7884** |

Notes.
The best FID are shown in bold.

AR dataset, 60 under the XM2VTS dataset, and 25 under the SKSF-A dataset. Table 3 shows that the complete model performs best in LP, while maintaining good CF and GE.

Additionally, the synthetic image quality of each ablation experiment was lower than that of the overall model.

Removing the local sketch refinement module (LSRM) leads to the global generator lacking optimization constraints related to local refinement loss. Consequently, this results in poor synthesis of local details in the pseudo-sketches, as evidenced by significantly lower LP scores. For instance, as depicted by the images in Fig. 3C, the eyes and mouth areas suffer from severe blurring. Additionally, the removal of $L_1$ results in a lack of correlation between the local and global generators, reflected by fluctuations in the CE, GE, and LP metrics. There is a decrease in the quality of synthesized local features, as shown in the eye portion of the image with shadows in Fig. 3D. Removing $L_{GANP}$ and removing $L_{MBCP}$ result in insufficient local constraints on the global generator. It makes the generated pseudo-sketches poorly represented in the eyes, mouth, and nose parts, as reflected by significantly lower LP scores. The results of removing the "warmup-epoch" are shown in the four images corresponding to Fig. 3G. The absence of the "warmup-epoch" leads to a mismatch between local and global optimization, as evidenced by the inability to obtain better global results in the early stages. For example, the testing results under the AR dataset are severely missing at the end of the hair. Better local results could not be obtained later, such as the testing results under the CUHK and SKSF-A datasets with more severe shadows between the two lips. Additionally, the robustness of the network decreases, as evidenced by significant fluctuations in the FID, CF, GE, and LP metrics after each training session.

Based on the results presented in Fig. 3, Tables 2 and 3, it is evident that each component of the experimental design contributes to the enhanced sketch synthesis. The complete model yields a higher-quality synthesis effect compared to each individual ablation experiment.

## Comparison experiments

To verify the effectiveness and robustness of the HCGAN, this section compares it with existing methods, such as Cycle-GAN (*Zhu et al., 2017*) and pix2pix (*Lin, Pang & Xia, 2020*), among others. While FSGAN (*Fan et al., 2022*) adopts a similar global and local approach, Cycle-GAN and DRIT++ (*Lee et al., 2018*) use unpaired input, whereas pix2pix and FSGAN use paired input. Specifically, we also compare with the latest sketch face synthesis method, StyleSketch (*Yun et al., 2024*). The results for each dataset are presented in Figs. 4, 5, 6 and 7, and the quantitative metrics for each method on each dataset are shown in Tables 4 and 5.

Figures 4, 5, 6 and 7 show the comparison of the synthesis effect of each method under the three datasets, from left to right, for the optical image, the corresponding real sketch image, Cycle-GAN, pix2pix, DRIT++, FSGAN, HCGAN and StyleSketch respectively. As the figures show, Cycle-GAN relies on reconstruction to optimize the style transfer under unpaired input and obtains better global information. However, too many unnecessary pixel details are generated, and too much noise interferes, resulting in a less clear overall sketch image and poor visual perception. When processing the AR dataset, the eyes were severely deformed in the synthetic results. The pix2pix method relies on conditional GANs for sketch synthesis, aided by $L_1$ loss optimization. However, the synthesized sketches are

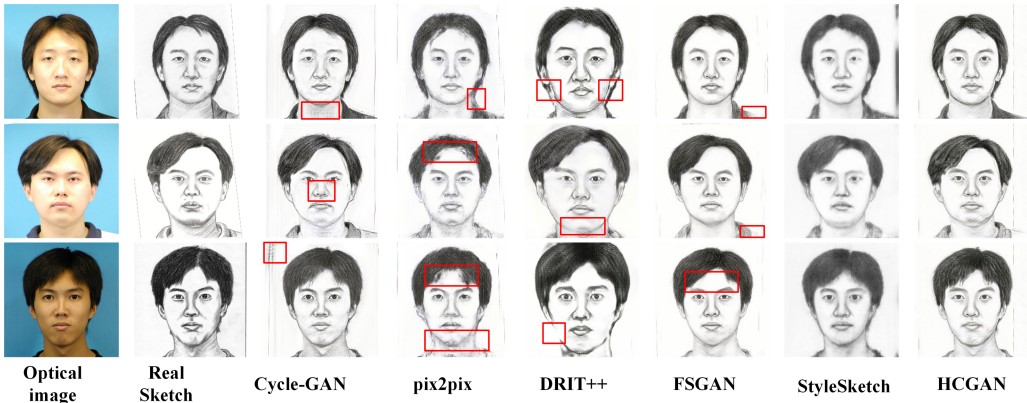

| Optical image | Real Sketch | Cycle-GAN | pix2pix | DRIT++ | FSGAN | StyleSketch | HCGAN |

**Figure 4** Comparison of synthesis effects of different methods on the CUHK dataset.

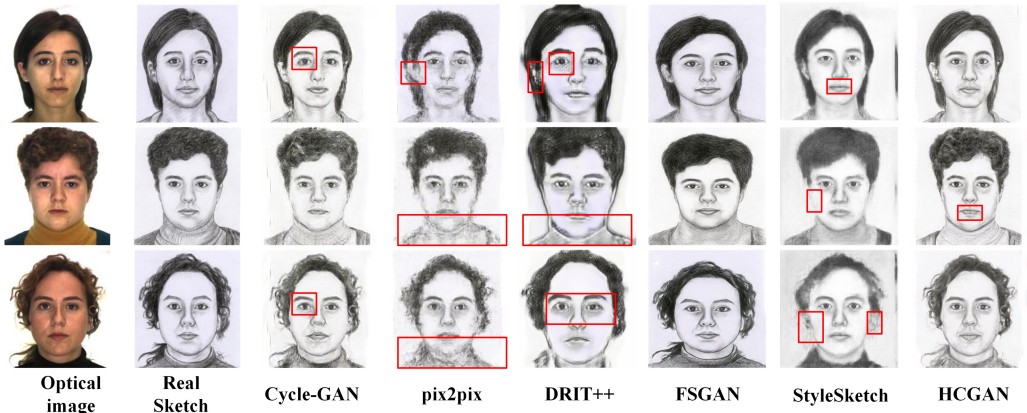

| Optical image | Real Sketch | Cycle-GAN | pix2pix | DRIT++ | FSGAN | StyleSketch | HCGAN |

**Figure 5** Comparison of synthesis effects of different methods on the AR dataset.

severely distorted and the results are unsatisfactory. The main reason is that optical and real sketches are not strictly pixel-aligned, and using $L_1$ loss leads to shifting and blurring key details. DRIT++ is a general method for style transfer, but it suffers from the problem of contour ghosting in composite sketches. In the XM2VTS dataset, partial distortion exists in the synthesized sketches' eyes despite the overall improvement in results. FSGAN splits and migrates the local part before fusion, creating a clear local portrayal. However, when processing XM2VTS, synthetic images have heavy shadows at the junction of local and other parts, and the double chin details are lost. Additionally, clothing edges lack global aspects, and the FID of synthetic sketches is poor. StyleSketch performs well on the CUHK and SKSF-A datasets but performs poorly on the AR and XM2VTS datasets. The fundamental reason lies in its non-end-to-end architecture, where the quality of synthesized results relies on the effectiveness of the encoder4editing (*Tov et al., 2021*) in extracting latent code. When the extracted latent code fails to capture the content of optical images effectively,

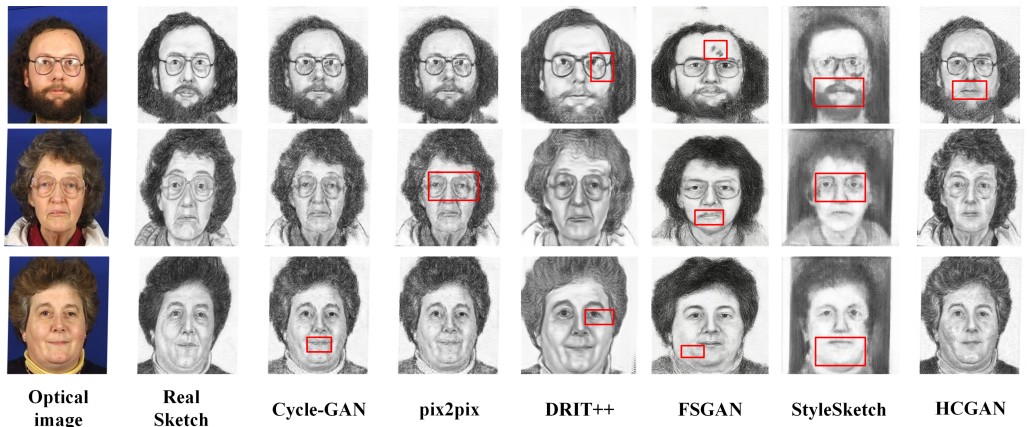

**Optical image** | **Real Sketch** | **Cycle-GAN** | **pix2pix** | **DRIT++** | **FSGAN** | **StyleSketch** | **HCGAN**

**Figure 6**  Comparison of synthesis effects of different methods on the XM2TVS dataset.

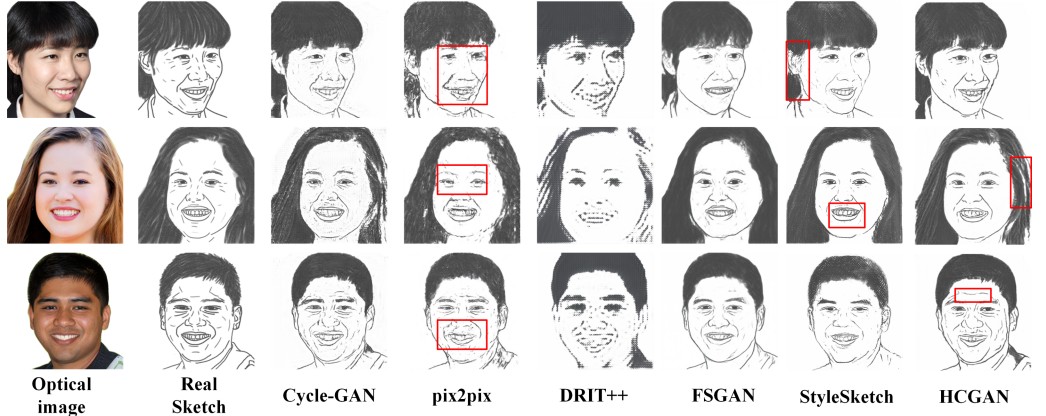

**Optical image** | **Real Sketch** | **Cycle-GAN** | **pix2pix** | **DRIT++** | **FSGAN** | **StyleSketch** | **HCGAN**

**Figure 7**  Comparison of synthesis effects of different methods on the SKSF-A dataset.

**Table 4**  Fréchet inception distance of different methods.

|  | Cycle-GAN | pix2pix | DRIT++ | FSGAN | StyleSktech | HCGAN |
|---|---|---|---|---|---|---|
| CUHK | 72.6964 | 110.7320 | 72.9310 | 61.2710 | 82.6866 | **48.5769** |
| AR | 70.1536 | 105.2783 | 82.5112 | 66.9377 | 105.5324 | **62.0253** |
| XM2VTS | 41.5939 | 72.4432 | 55.8660 | 39.6640 | 200.7224 | **30.6324** |
| SKSF-A | 118.3368 | 123.8700 | 117.5661 | 93.9339 | 88.1855 | **80.7176** |

**Notes.**
The best FID are shown in bold.

severe blurring occurs in the synthesized images. Additionally, the CUFS weights provided by StyleSketch are ineffective in eliminating the influence of the background.

Tables 4 and 5 presents the FID, CF, GE and LP metrics of different methods on different datasets, where the best results are highlighted in bold. It can be observed that HCGAN in

**Table 5  CF, GE, and LP of different methods.**

|        | Cycle-GAN | pix2pix | DRIT++ | FSGAN  | StyleSktech | HCGAN      |
|--------|-----------|---------|--------|--------|-------------|------------|
| CUHK   | 0.6031    | 0.6034  | 0.5944 | 0.6222 | 0.6022      | **0.7675** |
|        | 0.7741    | 0.8088  | 0.8039 | 0.8023 | 0.7940      | **0.8168** |
|        | 0.6822    | 0.6984  | 0.6957 | 0.6735 | 0.6957      | **0.7597** |
| AR     | 0.6085    | 0.5788  | 0.5923 | 0.6159 | 0.5978      | **0.8235** |
|        | 0.8578    | 0.8388  | 0.7982 | 0.8071 | 0.8675      | **0.8691** |
|        | 0.6650    | 0.6576  | 0.6482 | 0.6848 | 0.6236      | **0.7815** |
| XM2VTS | 0.5280    | 0.4977  | 0.5254 | 0.5173 | 0.4213      | **0.7190** |
|        | 0.9061    | 0.9074  | 0.7609 | 0.8633 | 0.7038      | **0.9538** |
|        | 0.6465    | 0.6369  | 0.6459 | 0.6418 | 0.5934      | **0.7457** |
| SKSF-A | 0.4577    | 0.4622  | 0.4035 | 0.4900 | 0.6851      | **0.7230** |
|        | 0.9213    | 0.9602  | 0.6354 | 0.9610 | 0.9332      | **0.9783** |
|        | 0.6343    | 0.6407  | 0.4012 | 0.6449 | 0.6865      | **0.7883** |

**Notes.**
The best indicators are shown in bold.

this article outperforms other comparative methods in terms of FID values across all three datasets, achieving the best performance in terms of content, global, and local aspects.

It is worth mentioning that the ability of HCGAN to refine local details depends on the capability of the local sketch refinement module to extract key local features. From Figs. 5 and 6, it can be observed that there is some blurriness in the mouth area of the synthesized images by HCGAN. The most likely reason for this is the ineffective extraction of key local features, leading to a decrease in the effectiveness of the local refinement loss. Figure 7 shows that the details in the hairline and forehead areas of the sketches synthesized by HCGAN on the SKSF-A dataset are still lacking. This inadequacy maybe since the SKSF-A dataset includes not only frontal faces but also profiles, which require more emphasis on the global-level portrayal. Under similar dataset scales, maintaining the same CUHK and other dataset settings may result in a deficiency in global-level portrayal.

Based on visual comparison and quantitative evaluation, our proposed method outperforms many existing image-to-image translation methods that use unpaired data. Specifically, our method generates synthesized sketches with more accurate and detailed local features. Whether it is the CUHK and AR datasets with fewer accessories, the XM2VTS dataset with more accessories, or the SKSF-A dataset with large angle differences, HCGAN achieves the best synthesis results, reflecting its effectiveness and robustness.

## CONCLUSION

HCGAN effectively solves the problems of relying on paired data training and local detail distortion faced by the current field of sketch face synthesis. It consists of two modules: a global sketch synthesis module and a local sketch refinement module. The global sketch synthesis module differs from Cycle-GAN because it uses a global multi-layer block contrast loss to enhance the relationship between input and output. The local sketch refinement module with good feature extraction capabilities provides local refinement loss to optimize the local details of the synthetic sketch. The local consistency loss and "warmup-epoch" strategy ensure efficient optimization of both modules. These improvements have led

HCGAN to outperform other methods on multiple datasets significantly. However, our method also has some limitations. For example, the effectiveness of the proposed local refinement loss is limited by the extraction of local features. Additionally, under small datasets and complex angles, the synthesis effectiveness at the global-level remains inadequate. In the future, the field of sketch face synthesis should pay more attention to depicting local details under unpaired training. In the future, a model that better guarantees global coordination and a more flexible and effective local loss should be proposed.

### Funding
This work was supported by the National Natural Science Foundation of China(No.62201066, No.62001033, No.U20A20163). In the study design phase, external funders provided input on the research questions and objectives. In the analysis phase, external funders collaborated with the research team to interpret the results. The decision to publish was a joint effort, with input from both the researchers and the funders. The manuscript preparation involved feedback and suggestions from the external funders, who also reviewed the final draft before submission.

### Grant Disclosures
The following grant information was disclosed by the authors:
The National Natural Science Foundation of China: No. 62201066, No. 62001033, No. U20A20163.

### Competing Interests
The authors declare there are no competing interests.

### Author Contributions
- Kangning Du conceived and designed the experiments, performed the experiments, performed the computation work, authored or reviewed drafts of the article, and approved the final draft.
- Zhen Wang conceived and designed the experiments, performed the experiments, performed the computation work, prepared figures and/or tables, authored or reviewed drafts of the article, and approved the final draft.
- Lin Cao conceived and designed the experiments, analyzed the data, performed the computation work, authored or reviewed drafts of the article, and approved the final draft.
- Yanan Guo conceived and designed the experiments, performed the experiments, analyzed the data, performed the computation work, prepared figures and/or tables, authored or reviewed drafts of the article, and approved the final draft.
- Shu Tian conceived and designed the experiments, analyzed the data, prepared figures and/or tables, authored or reviewed drafts of the article, and approved the final draft.
- Fan Zhang conceived and designed the experiments, analyzed the data, prepared figures and/or tables, authored or reviewed drafts of the article, and approved the final draft.

## Data Availability

The processed data is available at figshare: Zhen, Wang (2024). The dataset used in the article "HCGAN: Hierarchical Contrast Generative Adversarial Network for Unpaired Sketch Face Synthesis". figshare. Dataset. https://doi.org/10.6084/m9.figshare.26074954.v1.

The CUFS dataset is available at DOI https://doi.org/10.1109/TPAMI.2008.222 and also available at https://www.kaggle.com/datasets/arbazkhan971/cuhk-face-sketch-database-cufs.

The SKSF-A dataset is available at GitHub: https://github.com/kwanyun/SKSF-A/; from doi: 10.1111/cgf.15045.

Before experimentation, the face images and corresponding sketches from the datasets above underwent alignment through the Multi-task Cascaded Convolutional Neural Networks (MTCNN) algorithm. They were then uniformly cropped to 256 × 256 pixels size. Dilb was used to collect facial points to assist with face cropping. In addition, the relevant training parameters should be specified during training and testing. The processed data set and training and testing scripts are provided for convenience.

## Supplemental Information

Supplemental information for this article can be found online at http://dx.doi.org/10.7717/peerj-cs.2184#supplemental-information.

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
