# Peer review of "HCGAN: hierarchical contrast generative adversarial network for unpaired sketch face synthesis"

_PeerJ Computer Science, doi:10.7717/peerj-cs.2184_

## Round 0.1 · original submission · Major Revisions

The reviewers acknowledge the relevance of the manuscript. However, they underline to use more metrics to assess the performance of the proposed method and to provide an ablation study to investigate the effectiveness of the proposal.

**Language Note:** The review process has identified that the English language must be improved. PeerJ can provide language editing services - please contact us at [email protected] for pricing (be sure to provide your manuscript number and title). Alternatively, you should make your own arrangements to improve the language quality and provide details in your response letter. – PeerJ Staff

Reviewer 1 ·

Basic reporting

1. Incorporate specific result details within the abstract for a comprehensive overview.
2. Enhance the related work section by either incorporating a summarizing paragraph or integrating a comparison table to provide a condensed comparison of various methodologies.
3. Emphasize the challenges associated with deep learning techniques mentioned in the paper.

Experimental design

4. Address the choice of utilizing a dataset from 2008, explaining the rationale behind opting for an older dataset.
5. Acknowledge the limitation of the dataset's size and propose an expansion to enhance the robustness of the study. The authors should increase the dataset size and rerun the experiments.

Validity of the findings

6. Establish a dedicated section for evaluation metrics, elucidating each metric before its application in Table 1.
7. Augment the experiment section by including specific hardware and parameter details for transparency and reproducibility.
8. Conclude the paper by not only summarizing findings but also delving into potential future research directions and acknowledging current technique limitations.
9. Enrich the paper with recent references to ensure the inclusion of the latest advancements and developments in the field.

Reviewer 2 ·

Basic reporting

Please see my detailed comments.

Experimental design

Please see my detailed comments.

Validity of the findings

Please see my detailed comments.

Additional comments

This paper investigated a method for face synthesis. The idea sounds clear and the applications look promising. Authors provided certain convincing results for the verification of the proposed method. To further improve the paper, I have the following comments:

1) some equations should be indexed in the main context or desribed appropriately.

2) in experiment, authors should give some failure cases associated with discussions for the benifit of future work.

3) now diffusion models are quite popular in the generative AI field. Why do we use GAN for this task, instead of diffusion models, which should give stable and more promising results in this task.

Reviewer 3 ·

Basic reporting

The paper structure is organized well. It is clear to understand.
For the loss function, the authors should explain weights. If I set the same value for all weights, how performance will be?

Experimental design

The authors must expand the experiments. More test cases should be considered. Moreover, the quality metric "Frechet Inception Distance" also need to be added to the paper.

Validity of the findings

The paper proposed a novel method based on GAN to generate sketch effect from an input image. Basically, the resulted image relates to artistic effects. I recommend the authors find other metrics to assess the final result, not only the distance metric (Frechet). The authors are recommended reading this paper: [Evaluate and improve the quality of neural style transfer, Computer Vision and Image Understanding]

Additional comments

- All symbols and notations should be explained.
- English text also needs to be improved. There are some typos and grammar mistakes.

---

## Round 0.2 · accepted · Accept

Thank you for submitting to PeerJ Computer Science Journal. The authors have reviewed the manuscript according to the reviewers' suggestions.